# Organic Fouling Impact in a Direct Contact Membrane Distillation System Treating Wastewater: Experimental Observations and Modeling Approach

**DOI:** 10.3390/membranes11070493

**Published:** 2021-06-30

**Authors:** Amine Charfi, Fida Tibi, Jeonghwan Kim, Jin Hur, Jinwoo Cho

**Affiliations:** 1Department of Environment & Energy, Sejong University, Seoul 05006, Korea; amine.charfi@ymail.com (A.C.); jinhur@sejong.ac.kr (J.H.); 2Program of Environmental and Polymer Engineering, Department of Environmental Engineering, Inha University, Michuholgu, Inharo 100, Incheon 22212, Korea; tibifida7@gmail.com (F.T.); jeonghwankim@inha.ac.kr (J.K.)

**Keywords:** membrane distillation, organic fouling, wastewater, sodium alginate, bovine serum albumin, modeling

## Abstract

This study aims to investigate the effect of operational conditions on organic fouling occurring in a direct contact membrane distillation (DCMD) system used to treat wastewater. A mixed solution of sodium alginate (SA) and bovine serum albumin (BSA) was used as a feed solution to simulate polysaccharides and proteins, respectively, assumed as the main organic foulants. The permeate flux was observed at two feed temperatures 35 and 50 °C, as well as three feed solution pH 4, 6, and 8. Higher permeate flux was observed for higher feed temperature, which allows higher vapor pressure. At higher pH, a smaller particle size was detected with lower permeate flux. A mathematical model based on mass balance was developed to simulate permeate flux with time by assuming (i) the cake formation controlled by attachment and detachment of foulant materials and (ii) the increase in specific cake resistance, the function of the cake porosity, as the main mechanisms controlling membrane fouling to investigate the fouling mechanism responsible of permeate flux decline. The model fitted well with the experimental data with R^2^ superior to 0.9. High specific cake resistance fostered by small particle size would be responsible for the low permeate flux observed at pH 8.

## 1. Introduction

Membrane distillation (MD) is a thermally driven membrane process able to separate components according to their volatility. Thus, the MD system retains contaminants with low volatility and allows the production of highly purified water [1]. The driving force in the MD system is the difference in the vapor pressure between the feed and the permeate sides, which is related to the difference in temperature between the two compartments. As the temperature in the feed solution increases, the more volatile component would evaporate, cross a hydrophobic membrane, and condensate on the permeate side. Nowadays, the MD system is regarded with great interest for wastewater treatment due to its high separation effectiveness, low energy consumption, and low fouling risk compared to the pressure-driven membrane separation processes. Several studies showed the high performance of the MD system applied for wastewater treatment [2,3,4,5,6,7]. Nevertheless, even if operated at low pressure, fouling still occurs and causes a decline in MD performance. Membrane fouling leads to permeate flux decrease and consequently reduces the MD system productivity. In some cases, the fouling layer could act as a dynamic secondary membrane which can increase the rejection performance of the system. Nevertheless, for long-term operation, the fouling would promote membrane wetting which has an adverse effect on rejection performance [4], and it should be then controlled for an optimal operation of the MD process [8,9]. Membrane fouling is due to the deposit of organic and inorganic matter onto the membrane surface and/or within its pores, leading to the deterioration of the process performances [10]. Fouling occurring in the MD system could develop a heat resistance which promotes the temperature polarization phenomenon and consequently decreases the vapor pressure difference and the mass transfer [11,12]. Moreover, fouling could decrease the mass transfer by reducing the membrane permeability. Several methods have been studied to limit fouling in MD, such as feed solution pretreatment [13,14,15], physical/chemical cleaning [16], and membrane material modifications [17,18]. The modeling approach would be an effective way to better understand fouling mechanisms and to assess their effect on membrane productivity to control the fouling phenomenon. Nevertheless, few attempts at fouling modeling have been made for the MD system. Banat et al. [19] developed a model to simulate the effect of the temperature and concentration polarization on permeate flux decline. Warsinger et al. [20] developed a modeling approach based on thermodynamics, fluid mechanics, and kinetics to predict the scaling development in the MD system without including its effect on the flux decline. Even if developed for the pressure-driven membrane separation process, several studies adopted the pore blocking models [21] to identify the fouling mechanisms in the MD system [22,23]. Ramezanianpour and Sivakumar [24] combined the effects of intermediate blocking and cake formation to simulate the flux decline. The accuracy of those models to assess the effect of membrane fouling on flux decline in the MD system is still limited.

In this study, a mathematical model was developed to assess the effect of cake deposits on permeate flux decline in a direct contact membrane distillation (DCMD) system. This model emphasizes the effect of the deposit structure through its specific cake resistance, according to the feed solution pH and temperature. Moreover, organic fouling has been observed to affect the MD process productivity significantly [25]. Thus, experiments were conducted using synthetic solutions including both Sodium alginate (SA) and bovine serum albumin (BSA) to simulate polysaccharides and proteins, respectively, which are usually identified as the main organic foulants [26,27,28]. The obtained data were used to validate the developed model. Even if membrane fouling could be affected by numerous parameters related to the feed solution characteristics, membrane properties, and process configuration [17], this study will focus only on the effect of the feed solution pH and temperature, which significance was proven by previous works [4,23].

## 2. Materials and Methods

### 2.1. Experimental Setup

The experiments were conducted using a DCMD system [18]. The schematic of the MD setup is available in Annex 1. A commercial hydrophobic Polyvinylidene fluoride (PVDF) flat sheet membrane with a total area of 0.01 m^2^ placed inside the module with the pore size of 0.22 µm and a contact angle of 111°, was used. The feed and permeate circulations were maintained at 500 mL min^−1^ using gear pumps (WT3000-1FA, Longer pump, Baoding, China). The feed solution was heated by a thermostat water bath (HS-DBW-11, Digital water bath, Hansol, Seoul, Korea). Two different feed temperatures, including 35 and 50 °C, were applied in this study. The temperature of permeate side was maintained at 25 °C. The permeate was weighed by a balance, and the weight data were logged into the computer every 15 min for 24 h. Sodium alginate (SA, Duksan Pure Chemicals, Ansan, Korea) and bovine serum albumin (BSA, Amresco, Cleveland, OH, USA) were selected as model foulant materials in this study. The feed solution was prepared by dissolving 350 mg/L SA and BSA mixture in deionized (DI) water. Three different solutions, pH 4, 6, and 8, were tested. The pH was adjusted by adding hydrochloric acid (0.1 M) or sodium hydroxide (0.1 M). Cross-sectional images were taken using scanning electron microscopy (SEM) to observe membrane fouling. The particle size analysis was also carried out by using a particle size analyzer (Otsuka Electronics, ELS-Z, Japan).

### 2.2. Modelling Approach

According to the SEM images, the proposed model assumed the membrane fouling by cake formation as the main responsible for the flux decline. Thus, the permeate flux was expressed by resistance in a series model (Equation (1)), as a ratio of (i) the difference in the vapor pressure between the feed and permeate sides of the membrane, and (ii) the total resistance, which is the sum of the cake resistance and the intrinsic membrane resistance.
(1)$$J=\frac{\Delta P}{R_{T}}=\frac{\Delta P}{R_{m}+R_{c}}\;$$
with J the permeate flux (m^3^·m^−2^·s^−1^), ΔP the difference in the vapor pressure (Pa), RT the total membrane resistance (m^−1^·Pa·s), R_m_ the intrinsic membrane resistance (m^−1^·Pa·s), and R_c_ the cake resistance (m^−1^·Pa·s).

In this study, the difference in vapor pressure depended only on the feed solution characteristics and was not affected by the developed fouling layer. It was then assumed constant with time. This assumption was based on the fact that while treating wastewaters with the MD system, the fouling layer developed with high water content was characterized by high thermal conductivity [11], which makes its effect on heat transfer and consequently on temperature polarization negligible. Thus, the effect of the fouling layer development on vapor pressure was also assumed to be negligible. In each studied operational condition, the difference in vapor pressure was calculated based on the experimental value of the initial permeate flux J_0_, using Equation (2).
(2)$$\Delta P={\;R}_{m}\cdot J_{0}$$
with R_m_ the intrinsic membrane resistance had a constant value of 2.95 × 10^9^ (m^−1^·Pa·s), which is a characteristic of the commercial membrane used in this study.

The cake resistance was assumed as proportional to the viscosity of the water crossing the cake layer, the cake mass, and the specific cake resistance, as expressed in Equation (3). According to the feed solution temperature, different water viscosity values were used, namely 0.72 m·Pa·s at 35 °C and 0.54 m·Pa·s at 50 °C.
(3)$$R_{c}=\frac{µ\;\cdot \alpha \cdot m_{c}}{A}$$
with µ the water viscosity (Pa·s), α the specific cake resistance (m·kg^−1^), m_c_ the cake mass (kg) and A the membrane area (m^2^).

The cake mass is the solution of the differential equation expressed in Equation (4), showing that the variation of the cake mass is equal to the difference in the mass of matter attached to the membrane surface (Equation (5)) and the matter detached from the membrane surface (Equation (6)) by turbulence effect created by the feed solution recirculation.
(4)$$\frac{{\mathrm{dm}}_{c}}{\mathrm{dt}}=\frac{{\mathrm{dm}}_{a}}{\mathrm{dt}}-\frac{{\mathrm{dm}}_{d}}{\mathrm{dt}}$$
(5)$$\frac{{\mathrm{dm}}_{a}}{\mathrm{dt}}=k_{a}C_{\mathrm{SA},\mathrm{BSA}}$$
(6)$$\frac{{\mathrm{dm}}_{d}}{\mathrm{dt}}=k_{d}m_{c}$$
with m_a_ the mass of matter attached to the membrane surface (kg), m_d_ the mass of detached matter (kg), k_a_ the kinetic parameter of matter attachment to the membrane surface (m^3^·s^−1^), C_SA, BSA_ the concentration of the mixed SA and BSA feed solution (kg·m^−3^) and k_d_ the kinetic parameter of matter detachment from the membrane surface (s^−1^).

The specific cake resistance was determined using the Kozeny–Carman’s relation (Equation (7)),
(7)$$\alpha =\frac{180\;\left(1-\varepsilon \right)}{\rho_{s}\cdot d^{2}\cdot \varepsilon^{3}}$$
with ε the cake porosity, ρs cake density (kg·m^−3^), d the cake particle mean diameter (m).

The model assumes that the cake porosity ε would decrease with time according to a first-order differential equation (Equation (8)).
(8)$$\frac{d\varepsilon }{\mathrm{dt}}=-k_{\varepsilon }\cdot m_{c}\cdot \varepsilon$$
with k_ε_ the parameter of the porosity dynamic (kg^−1^·s^−1^), m_c_ the cake mass (kg) and ε the cake porosity.

### 2.3. Model Validation and Parameters Identification Method

For model adjustment on permeate flux experimental data and model parameters’ identification, the least-squares method using MATLAB software was adopted. The method consisted of identifying the model parameters’ values allowing (i) the least-squares function FLS (Equation (9)) to be minimized, (ii) the correlation coefficient function R (Equation (10)) to be maximized, and (iii) the coefficient of determination function R^2^ (Equation (11)) to be maximized.
(9)$$F_{\mathrm{LS}}=\sum {\left(J_{\exp}-J_{\mathrm{sim}}\right)}^{2}$$
with J_exp_ the experimental data of permeate flux (m^3^·m^−2^·s^−1^) and J_sim_ the simulation data of permeate flux (m^3^·m^−2^·s^−1^).
(10)$$R=\frac{\mathrm{cov}\left(J_{\exp},J_{\mathrm{sim}}\right)}{\sqrt{\mathrm{var}\left(J_{\exp}\right)\cdot \mathrm{var}\left(J_{\mathrm{sim}}\right)}}$$
with cov(J_exp_, J_sim_) the covariance of J_exp_ and J_sim_, var(J_exp_), the variance of J_exp_ and var(J_sim_) the variance of J_sim_.
(11)$$R^{2}=1-\frac{\sum {\left(J_{\exp}-J_{\mathrm{sim}}\right)}^{2}}{\sum {\left(J_{\exp}-\bar{J_{\exp}}\right)}^{2}}$$
with $\bar{J_{\exp}}$ the average value of the experimental values of permeate flux.

The efficiency of the model fitting with experimental data was evaluated by calculating the coefficient of determination R^2^.

## 3. Results and Discussion

### 3.1. Effect of Feed Solution pH and Temperature on Permeate Flux Decline

The permeate flux observed during the distillation of SA-BSA solutions at different pH 4, 6, and 8, and temperatures 35 °C and 50 °C, are shown in Figure 1. For all studied conditions, the permeate flux decreased in the first phase due to the membrane fouling, then kept a constant value during a second phase until the end of the experiment, which shows that the fouling layer developed kept constant characteristics during the second phase. The fouling layer developed is mainly due to ionic interactions between alginate and BSA molecules [29]. Higher permeate flux was observed for higher temperatures. In fact, the vapor pressure difference, which is the driving force in the MD system, was higher at a higher feed solution temperature. Regardless of the feed solution temperature, lower permeate flux was observed at higher pH. While the interactions between alginate and BSA were higher at pH 4 and promoted the formation of aggregations, at pH 6 and 8, alginate-BSA soluble complexes were formed. The complexes alginate-BSA have a lower size and are more soluble as pH increases [30]. As water evaporates, soluble complexes reconcentrate in the vicinity of the membrane surface and lead to a concentration polarization phenomenon which could evolve to form a deposit by chain entanglements [29]. Thus, the fouling layer formed at higher pH would be denser, which explains the lower permeate flux observed at pH 8.

SEM images of the membrane cross-section taken for each studied case are displayed in Figure 2. In all studied conditions, the SEM images showed a deposit formed on the membrane surface which was responsible for the flux decline. The deposit layer seemed thicker at high feed temperature for all studied pH. At higher feed temperature, the higher vapor pressure difference created allowed higher convective forces able to drag higher amounts of foulants to the membrane surface. This would promote a rapid accumulation of a high amount of foulant, leading to the building of a thicker fouling layer. However, at low feed temperature, lower convective forces toward the membrane were created. Thus, the lower amount of foulants were dragged slowly towards the membrane surface, leading to the building of a thinner fouling layer. At high feed temperature, lower deposit thickness was observed when pH was increased, which could be explained by the effect of acidic conditions in promoting SA and BSA aggregation [31,32]. Nevertheless, at similar feed temperature, the opposite trend was observed by Yan et al. [23], who showed thicker deposits when increasing pH from 5 to 8. This observation was due to the effect of scaling, which was not considered in our study. The scaling occurring at high pH fosters the organic fouling, as organics could be attached to the scaling layer [23].

### 3.2. Particle Size Distribution

The particle size distribution is an important parameter in characterizing the fouling layer structure, as shown by the Kozeny–Carman relation (Equation (7)), which expresses the specific cake resistance in the function of the particle size. The particle size distribution in mixed SA-BSA solution obtained at pH 4, 6, and 8 is shown in Figure 3. In all three conditions, two peaks are shown, one high and one low peak. The high peak corresponds to the more dominant particle size. At pH 4, the high peak was observed at a diameter of 16,200 nm and the low peak at a diameter of 1021 nm. At pH 4, large SA-BSA complexes were formed due to the electrostatic attraction between the positively charged BSA and the negatively charged SA. Harnsilawat et al. [31] also observed large complexes with sizes superior to 1000 nm at pH 4. Moreover, around its isoelectric point (4.9), BSA tended to aggregate due to hydrophobic attraction, van der Waals attraction, or electrostatic attraction between positively charged groups in some proteins and negatively charged groups in others. Furthermore, the protonation of carboxylic groups in the alginate structure leads to the formation of alginic acids, which tend to precipitate [32,33]. In addition, entanglements between alginate and protein chains could help the formation of bigger aggregates [29,31] and this phenomenon could be enhanced by the feed solution reconcentration due to its evaporation nearby the membrane surface in the MD system.

At pH superior to 5, both SA and BSA are negatively charged, thus at pH 6 and pH 8, the electrostatic repulsion created between negatively charged SA and BSA would hinder SA-BSA complexes formation [31]. This phenomenon could explain the peaks at 24.9 nm and 18 nm observed at pH 6 and pH 8, respectively, which could correspond to either BSA or SA. In fact, the BSA diameter was reported as having a range of 0.5 to 20 nm [30], and the SA diameter was reported by Motwani et al. [34] to be around 12 nm. Even if electrostatic repulsion was promoted at pH 6 and 8, high peaks were detected at 14,975 nm and 350 nm, respectively, showing that aggregates were formed probably by chain entanglement. The larger particle size detected at pH 6 than the one detected at pH 8 would be due to the lower negative charge of BSA and SA at pH 6, which decreased the electrostatic repulsion phenomenon. The difference in the particle size of SA-BSA solution observed at different pH was due to the significant effect of pH on SA and BSA charge and structure. In fact, in acidic conditions, high particle sizes observed were due to both SA and BSA aggregation and precipitate formation. However, at basic conditions, soluble and charged structures of SA and BSA were promoted which could explain the lower particle size observed.

### 3.3. Model Validation

For model validation, the simulations obtained by Equation (1) expressing the permeate flux were considered. The model simulations were fitted with the experimental data of permeate flux to validate the developed model and identify its parameters, namely, the attachment parameter k_a_, the detachment parameter k_d_, and the cake porosity parameter k_ε_. The cake particle mean diameter d values considered were 16.2, 14.9, and 0.35 µm (Table 1) for pH 4, 6, and 8, respectively. The considered values correspond to the most dominant particle diameter detected in the particle size distribution analysis conducted for SA-BSA mixed solution at different pH (Section 3.2). As shown in Figure 4, the model simulations fitted well with the experimental data with a coefficient of determination R^2^ superior to 0.90 and a correlation coefficient R superior to 0.91 for all studied pH and temperature conditions.

The model parameters were identified using the least-squares method detailed in Section 2.3. The values of least squares values obtained with the best fitting model are presented in Table 1. The model parameters’ values identified allowed the coefficient of determination R^2^ and the correlation coefficient R to be maximized. The attachment kinetic parameter k_a_ showed higher values at higher feed temperatures. In fact, higher feed temperatures allowed higher convective forces created by higher vapor pressure differences, which fosters matter drag to the membrane surface. Moreover, at high feed temperatures, higher water evaporation was allowed in the vicinity of the membrane surface, which promoted the concentration polarization phenomenon able in its turn to promote the matter attachment to the membrane material. The increase in SA and BSA concentration within the concentration polarization layer would reduce their negative charge and foster the chain entanglement phenomenon [29]. At both studied feed solution temperatures, the kinetic parameter of attachment showed its lowest value at pH 8. This observation could be explained by the fact that at pH 8, SA and BSA with high negative charge foster electrostatic repulsion between the SA and BSA molecules, which helped to keep their ionic soluble form and consequently promoted their dissolution in the feed solution and hindered their attachment to the membrane surface. In addition, the electrostatic repulsion between SA and BSA and membrane material negatively charged would hinder the cake formation. Similar results have been observed by Yan et al. [23] and Srisurichan et al. [35], who explained it by the increase in the membrane electronegativity at high feed solution pH as well as the high electronegativity of organic foulants, which hindered their adhesion to the membrane. The highest values of the parameter k_a_ were observed at pH 6. Even if the SA and BSA had ionic soluble forms at pH 6 [31], their high attachment tendency could be explained by the formation of aggregates by chain entanglement. This hypothesis could be supported by the particle size distribution results presented in Section 3.2, which showed the presence of large particle size at pH 6. The values of the detachment kinetic parameter k_d_ followed the same trend as the attachment parameter. This observation was expected since higher matter attached to the membrane surface allows a higher amount of matter detached by shear forces and back-diffusion mechanisms [36]. The values of the cake porosity dynamic parameter k_ε_ observed were generally low for all studied operational conditions showing the low effect of the mechanisms affecting the cake porosity decrease, such as the cake compressibility. Despite the compressible characteristic of SA and BSA [37], the low cake compression could be due to the low pressure applied in the membrane distillation system.

### 3.4. Specific Cake Resistance

The cake structure was investigated through the specific cake resistance expressed by the Kozeny–Carman relation (Equation (7)). The specific cake resistance simulation with time for different pH and feed temperatures is shown in Figure 5. The specific cake resistance estimated by the model for pH 4 and 6 showed low values compared to specific cake resistance estimated for pH 8, as 1.4 × 10^12^ m·kg^−1^ at 35 °C and 1.6 × 10^12^ m·kg^−1^ at 50 °C. This could be explained by the small particle size of the mixed SA-BSA solution detected at pH 8 (Section 3.2), which could form a denser cake layer. Moreover, higher specific cake resistance was estimated at higher feed solution temperatures. This observation could be explained by higher water evaporation near the membrane surface at higher temperatures, which promoted the concentration polarization phenomenon and could foster chain entanglement, leading to the formation of a denser cake layer. Higher specific cake resistance simulated at higher feed temperature could also be due to the effect of temperature increase on BSA structure. In fact, when temperature increased, a partial unfolding of BSA structure occurred, leading to the exposure of positively charged amino acids and hydrophobic nonpolar groups, such as alkyl and aromatic groups, which are usually observed inside the structure at atmospheric temperature [29,38]. This structure transformation would foster the electrostatic attraction between the negatively charged carboxyl groups of SA and positively charged groups of BSA. Moreover, it could promote the interaction between the hydrophobic groups of BSA and the hydrophobic membrane material. Both described mechanisms could increase the cake density and consequently the increase in the specific cake resistance. Moreover, in all studied cases, a slight increase in the specific cake resistance with time was observed. In fact, particles with low mean diameter detected in all feed solutions could enter and be trapped within the deposit pores leading to its porosity decline and consequently its specific cake resistance increase.

## 4. Conclusions

This study aimed to investigate the organic fouling responsible for the permeate flux decline in the DCMD system treating wastewater. SA and BSA were considered to simulate proteins and polysaccharides as the main organic foulants. The effect of the feed solution temperature and pH was studied, and lower permeate flux was observed when decreasing feed temperature to 35 °C and increasing pH to 8. To better understand the mechanisms that control organic fouling, a mathematical model was developed to simulate permeate flux with time by assuming the cake formation and the increase in specific cake resistance with time as the main aspects responsible for permeate flux decline. The model showed a satisfactory simulation accuracy of the experimental data with R^2^ superior to 0.9. The model simulation of specific cake resistance showed a slight increase with time due to the low pressure applied in DCMD and showed high values at pH 8 promoted by low SA-BSA particle size.

## Figures and Tables

**Figure 1 membranes-11-00493-f001:**
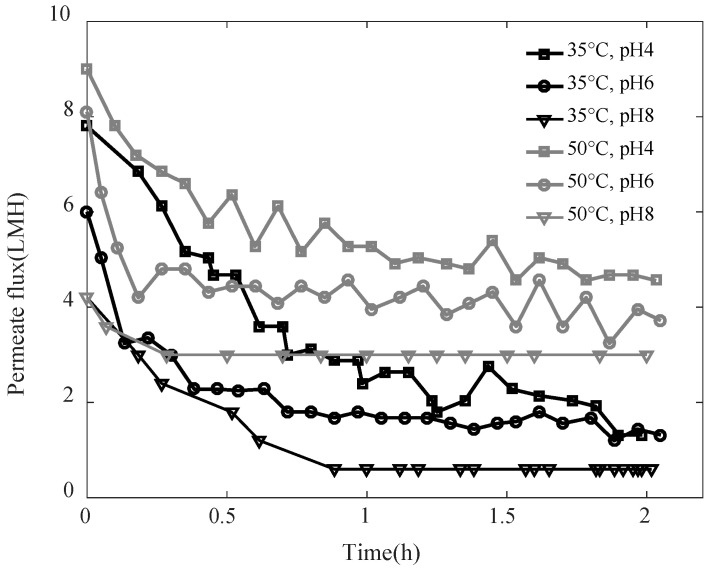
Permeate flux observed when distilling SA-BSA solution at different pH and temperatures.

**Figure 2 membranes-11-00493-f002:**
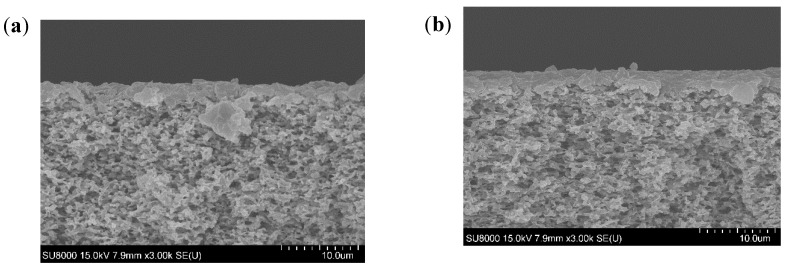
Cross-section view of the fouled membranes at different pH and temperatures (**a**) 35 °C, pH 4, (**b**) 35 °C, pH 6, (**c**) 35 °C, pH 8, (**d**) 50 °C, pH 4, (**e**) 50 °C, pH 6, and (**f**) 50 °C, pH 8.

**Figure 3 membranes-11-00493-f003:**
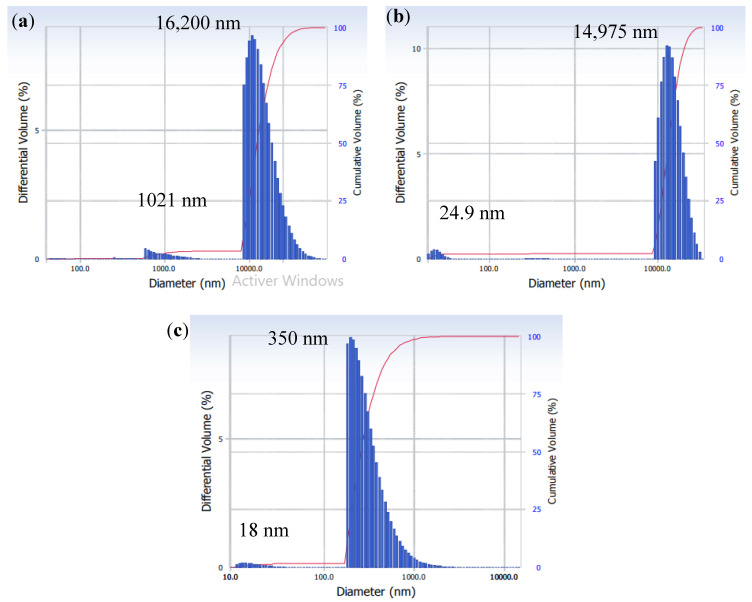
Particle size distribution of SA-BSA solution at (**a**) pH 4, (**b**) pH 6, and (**c**) pH 8.

**Figure 4 membranes-11-00493-f004:**
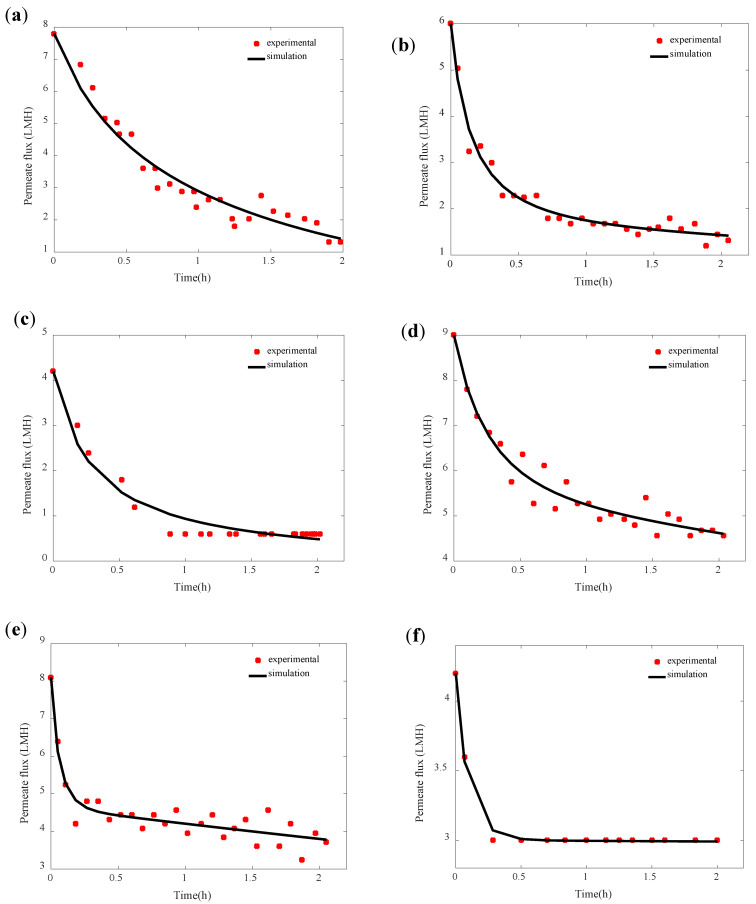
Comparison of the permeate flux experimental data and the model simulations obtained for different pH and temperatures (**a**) 35 °C, pH 4, (**b**) 35 °C, pH 6, (**c**) 35 °C, pH 8, (**d**) 50 °C, pH 4, (**e**) 50 °C, pH 6, and (**f**) 50 °C, pH 8.

**Figure 5 membranes-11-00493-f005:**
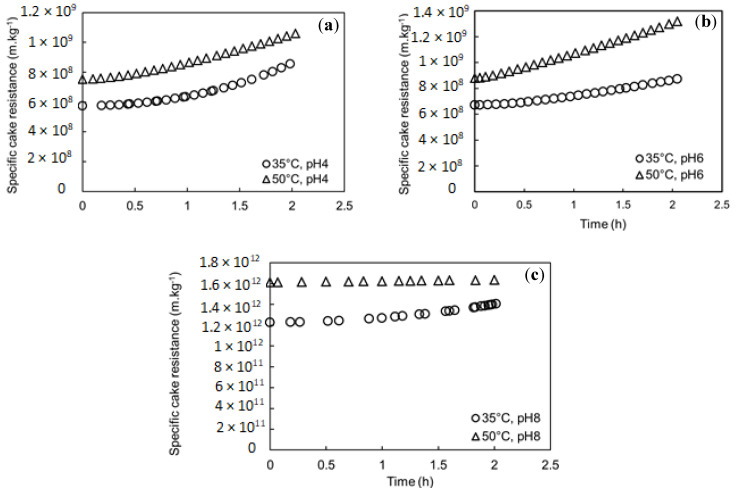
Simulations of Specific cake resistance with time at different feed temperatures and pH, (**a**) pH 4, (**b**) pH 6, and (**c**) pH 8.

**Table 1 membranes-11-00493-t001:** Identified values of the model parameters.

T (°C)	pH	d (µm)	k_a_(m^3^·s^−1^)	k_d_(s^−1^)	k_ε_(kg^−1^·s^−1^)	LS	R	R^2^
35	4	16.2	31.6896	0.001	0.0032	3.6527	0.9384	0.9498
	6	14.9	93.0179	2.0789	0.0019	0.7960	0.9496	0.9752
	8	0.35	0.0332	0.0100	0.9800	0.8164	0.9353	0.9584
50	4	16.2	34.6203	2.4372	0.0074	1.6387	0.9339	0.9463
	6	14.9	148.5366	10.9521	0.0077	2.2370	0.9134	0.9022
	8	0.35	0.0341	8.7230	0.9000	0.0069	0.9266	0.9957

## Data Availability

The data presented in this study are available on request from the corresponding author.

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
