# Peer review of "Organic Fouling Impact in a Direct Contact Membrane Distillation System Treating Wastewater: Experimental Observations and Modeling Approach"

_membranes, 2021, doi:10.3390/membranes11070493_

Round 1

Reviewer 1 Report

Dear Authors, 

I recommend major revision based on the following comments. 

1. For model validation, you used the least-squares function. There are other functions. Please, add at least two functions and compare them to your model validation. 

2.Please, illustrate the reason to focus on the effects of the feed solution pH and temperature only.

3.Please, state the consequences of the flux decline. 

5.The difference between the particle size of SA-BSA solution at pH 4 and other pH values is too big. What is the reason for that? 

6. Which equation did you conduct your model validation? in 3.3. Model validation: 

7. Conclusions are not covered to all points. Please, rewrite it. 

8. Please, insert the scheme of DCMD system as supplementary. 

Author Response

Dear Reviewer

please find attached detailed answers to your valuable comments

best regards

Jinwoo Cho

Reviewer 2 Report

Manuscript is written correctly. However, please see a few comments below:

  • There is no discussion comparing the results obtained by the Authors and the results available in the literature,
  • Lines 81, 122, 173, 208...: There is no space between numerals and degree symbol,
  • Line 182: There is no space between pH and 4,
  • The quality of the Table 1 should be improved,
  • The quality of the Figure 5 should be improved.

Author Response

Dear Reviewer

please find attached detailed responses to your valuable comments

best regards

Jinwoo Cho 

Round 2

Reviewer 1 Report

Dear authors, 

Thank you for the revised version of the manuscript.